# Land Use Quality Assessment and Exploration of the Driving Forces Based on Location: A Case Study in Luohe City, China

Xinyu Wang [1], Xinzhi Yao [2], Huamei Shao [3], Tian Bai [4], Yaqiong Xu [5], Guohang Tian [6], Albert Fekete [1] and László Kollányi [1,*]

1   Institute of Landscape Architecture, Urban Planning and Garden Art,
    Hungarian University of Agriculture and Life Sciences, 1114 Budapest, Hungary
2   China Construction Seventh Engineering Division Corp. Ltd., Zhengzhou 450004, China
3   Graduate School of Urban Studies, Hanyang University, Seoul 04763, Republic of Korea
4   College of Horticulture and Landscape, Yunnan Agricultural University, Kunming 650201, China
5   College of Environmental Science and Forestry, The State University of New York, New York, NY 13210, USA
6   College of Landscape Architecture and Art, Henan Agricultural University, Zhengzhou 450002, China
*   Correspondence: kollanyi.laszlo@uni-mate.hu; Tel.: +36-205753165

**Abstract:** With rapid urban population growth and industrial agglomeration, the urban land supply is becoming gradually tight. Improving land use quality (LUQ) is becoming increasingly critical. This study was carried out in the Luohe built-up zones between 2013 and 2021. The aim is to explore the growth characteristics of LUQ and determine the association between the inner urban location and the growth rate from the perspective of spatial heterogeneity. Therefore, based on a socio-economic-environmental framework, we selected an integration/GDP/population/artificial-surface Rate, and a remote-sensing-based ecological index to construct a LUQ assessment framework that is more stable and applicable for developing urban areas. Additionally, then, multiscale geographical weighted regression is adopted, which can better help us explore the scale of the location factors. The results show that: (1) The LUQ overall growth is gradually slowing. High-quality areas clustered in the urban center and subsystem elements spread outward along the national and provincial highways to drive boundary expansion; (2) In the W/E/SE direction, land use tends more towards physical sprawl than usual development and expansion; (3) Location factors were distinguished as global, semi-global, and local. The global factors constitute the homogenized locational space. Semi-global and local factors constitute a heterogeneous locational space. The latter is critical to guide LUQ growth. LUQ assessment can promote intensive land use. Exploring location factors can further guide the LUQ spatial growth and provide data in support of urban planning.

**Keywords:** quality assessment; driving mechanism; society-economy-environment; weighted regression; location theory

## 1. Introduction

The world, including China, has undergone rapid urbanization [1,2]. Urban boundaries are expanding dramatically, posing major challenges [3]. Promoting compact urban development and urban densification/re-utilization is often used against urban sprawl [4–6]. However, dense construction can also lead to adverse consequences, such as the compact city paradox [5,6]. Land use quality (LUQ) assessment can help people find the balance between the two, and can be of great significance to the planning, development and utilization of land resources more scientifically and rationally [7]. Currently, objective differences exist in the LUQ spatial distribution within cities. In some areas, densely built environments lead to the degradation of green space [6]. At the same time, there is also some idle construction land [8]. The LUQ of the same city can be very different in alternative locations. Therefore, it is necessary to study LUQ jointly with the spatial location in which it is located. The objective is to achieve LUQ assessment and to determine the association between locational

factors and growth. Controlling the key factors affecting growth is beneficial in promoting LUQ development and urban sustainability [9].

### 1.1. Land Use Quality Assessment

In China, LUQ is based on rational use as the goal, and according to the specific purpose, the quality appraisal of the attributes/efficiency/outputs of the land use is used to determine the value of the land [10]. The systematic cognition of elements, connotations, and dimensions is the basis of the assessment, which is mainly reflected in different frameworks: single and multi-objective frameworks [11].

The different land uses needed in different social stages give rise to different assessment frameworks. For example, when the urban is abstracted to a linear model, LUQ is defined as a single-index, such as economic [12] or physical [13] output. When the assessment gradually shifts from economics or taxation to rational use and sustainable development, people pay more attention to ecology, resources and social services. Additionally, along with urban multisystem research, such as physical and organic complex systems, urban areas exhibit the properties of naturally complex systems, and many mathematical models developed for studying naturally complex systems are applicable to urban areas [7,14–16]. The emergence of the synthetic framework reflects this shift in research concepts. It mainly includes socio-economic-environmental subsystems (Soc-Eco-Env) from the sustainable development field [17,18]. In urban areas, these three basic subsystems interact with each other [7,17,19]. Scholars also have combined, subdivided, or added extra [7,20,21] subsystems depending on the study area and objectives. In terms of study scale, not many studies have been conducted on the inner urban location, there are more studies about provinces [12,22], urban agglomerations [7,11], and cities [23,24], etc.

### 1.2. Urban Location Theory

Location theory [25–28] concerns how humans select locations for spatial activities. A typical criticism is that location theory is an economic model based on a spatial homogeneity assumption [29,30]. This is far from reality. The assumption of spatial homogeneity means that the city elements do not change with spatial location. Transit-oriented development theory states that urban growth is not homogeneous in all directions [31]. Although, over the next century, location theory evolved as a gradual relaxation of presuppositions [32]. Some theories add new dimensions to space by focusing on open space, employment, scale returns, market competition, and other factors [29,33,34]. However, these descriptions are relatively macroscopic and can hardly be used on an inner-urban scale. Studies have concentrated on the large scale of a few progressive cities, and need to pay more attention to changes in developing urban areas.

Natural conditions are the leading cause of urban formation [35,36]. Heterogeneity is gradually reinforced by road construction and local land aggregation [37]. The land driving forces for use vary significantly from country to country/region to region. Due to China's different institutional backgrounds and development stages, urban areas are driven by more macro influences, such as government planning [38]. In planning theory, the pole-axis system has profoundly impacted China's urban planning [39]. This theory emphasizes the heterogeneous influence of infrastructure, such as transportation and service facilities on the location [40]. Additionally, the sector model [41] also emphasizes land use and roads. The concentric circle model [42] emphasizes land use and the urban center location. Moreover, the multiple nuclei theory [28,43] emphasizes organization types, roads/stations, social service facilities, etc. According to the above discussion, the location factors are finally decomposed into three main parts: natural conditions, policy planning, and construction foundation.

Based on the land quality assessment, screening of location factors to analyze how they influence quality change. LUQ change intensity (LCI) is determined by calculating the annual rate of LUQ growth/decline, and it can help us better focus on quality change. Using multiscale geographical weighted regression (MGWR), the bandwidth results were analyzed to observe the impact pattern of location factors on LUQ, classified

into homogeneous and heterogeneous patterns. Homogeneous and heterogeneous location factors commonly influence LUQ growth, with the latter considered as a key factor of more significant concern [44].

## 2. Study Area and Materials

### 2.1. Flowchart

The roadmap is shown in Figure 1. The article includes three main parts: (1) Data set introduction and preliminary processing; (2) LUQ assessment and spatial analysis; (3) MGWR-based analysis of LCI locational driving mechanisms.

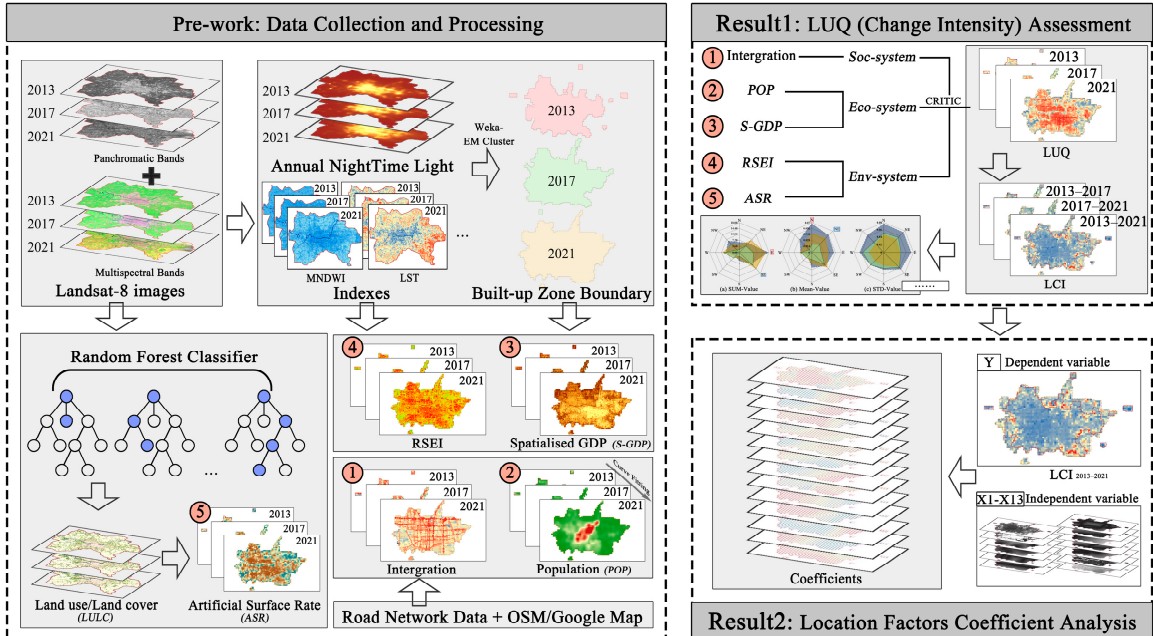

**Figure 1.** Flowchart: data preparation, LUQ assessment and MGWR calculation.

### 2.2. Study Area

Luohe (113°27′–114°16′ E, 33°24′–33°59′ N) is in the south-central part of Henan Province in central China and is generally flat. The Shali River cuts through it. There are three railways, two expressways, and one high-speed railway. The national/provincial highways intertwine here. Luohe is a regional transportation hub. In 2020, the population was 2.37 million (national ranking: 213/339). Similarly, the per capita gross domestic product (GDP) is slightly lower than the national mean (66.5 < 72.4, in thousand Yuan), but the growth rate is fast, with an average annual growth rate of about 9.1% in 2016–2020. It is a good representation of China's developing cities. The study years were 2013/2017/2021, and the study area was the built-up zones in the urban growth boundaries of Luohe (Figure 2).

The morphology of built-up zones is vital to judging the properties of the land system [45]. Reducing the data points outside the built-up zones can significantly improve the accuracy of the results. Luohe has not provided yet the built-up zones, so this study has used boundaries (Figure 2) obtained by clustering.

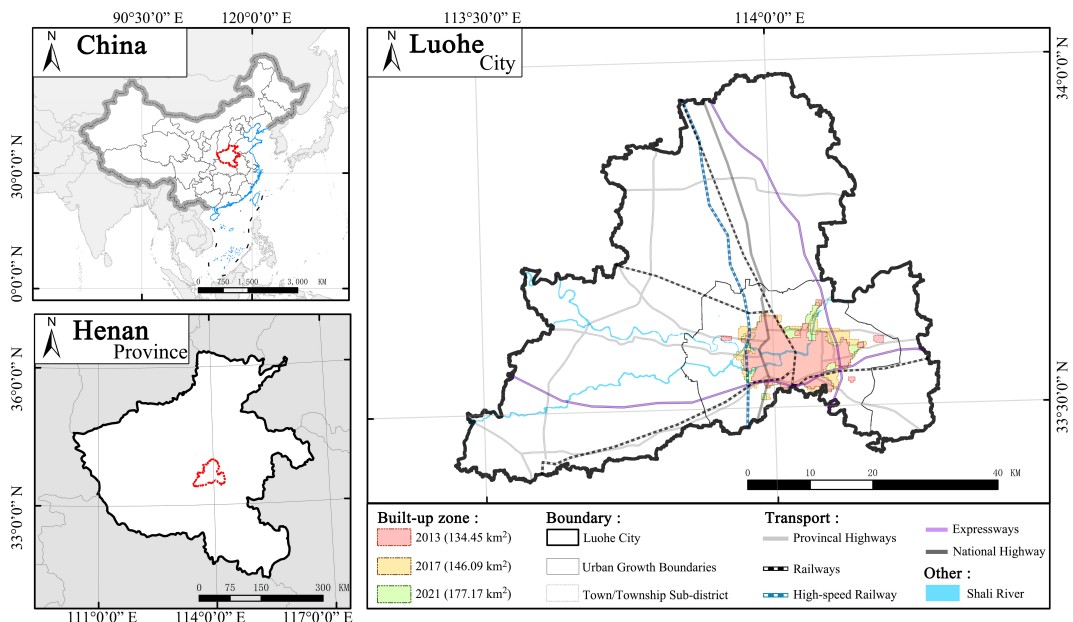

**Figure 2.** Location map (map of China from China Standard Map Service. GS (2020)4619).

### 2.3. Data Collection and Framework

#### 2.3.1. Data Collection and Processing

Satellite images were derived from Landsat-8 Surface Reflectance (SR) collection (https://earthexplorer.usgs.gov/, accessed on 1 June 2022). The annual nighttime lights (NTL) are from the NPP-VIIRS satellite (https://eogdata.mines.edu/nighttime_light/, accessed on 17 May 2022). Surface temperatures were produced using a split-window algorithm based on the thermal infrared (TIR) band [46,47].

DEM (Forest And Buildings removed Copernicus DEM) was used to describe the topography [48]. The roads were obtained from OSM (https://www.openstreetmap.org/, accessed on 1 June 2022) and have been manually corrected against Google Maps. Data on the location of development axes and regional cores were taken from planning documents. The 100 m population data were obtained from Worldpop (https://hub.worldpop.org/, accessed on 1 June 2022).

- Built-up zones and EM clustering

The identification of built-up zones draws on previous research in which the radial density of construction had a mutation threshold [49]. Expectation-maximum (EM) clustering is a type of curvilinear clustering that operates on a fixed feature vector and uses multiple iterations to make the likelihood value converge to an optimal solution [50]. Clustering was based on the NTL/normalized difference built-up index (NDBI)/ land surface temperature (LST) datasets. Urban growth boundaries have been classified into three clusters. In China, based on the traditional urban–rural dichotomy, there are also mixed urban–rural areas in developing urban areas, which is the "third area". In addition, the three-classification method can eliminate the blooming effects of NTL brightness.

- Land cover and random forest

Random forest is a machine-learning algorithm developed and refined by Leo Breiman and Adele Cutler in 2001. The method is widely used in land use classification studies in various cities in China [51] and worldwide [52]. The ArcGIS Pro platform calculated the normalized difference vegetation index (NDVI)/NDBI/modified normalized difference water index (MNDWI) as index features. Then, redundancy was removed from spectral bands by principal component analysis (PCA) to obtain three principal component bands. These were the spectral features. GLCM mean and GLCM variance were extracted as texture features based on the panchromatic band (B8) using the grey-level co-generation

matrix (GLCM). The above three features are fed into the random forest classifier to obtain the land use/land cover (LULC): 1 Farmland; 3 Water; 5 Green space; 7 Construction. The three kappa were 95.09%, 93.18%, and 96.36%.

### 2.3.2. LUQ Assessment Framework and Location Factor Selection

In the LUQ assessment framework, the social system provides the necessary infrastructure; the economic system provides the financial support to keep the entropy of the system low; the environmental system provides both the material basis and the ecological services for long-term development and progress [53,54]. Society selects the integration index. The spatialized GDP (S-GDP) [55] and population (POP) were chosen to represent the economy. The environment system uses the artificial surface rate (ASR) and remote-sensing-based ecological index (RSEI) to evaluate the grey-green matrix [11] (Table 1).

**Table 1.** LUQ Assessment framework and indicators.

| Target Layer | Framework Layer | Assessment Indicators |
|:---:|:---:|:---:|
| | Society | Integration |
| LUQ | Economy | Spatialized GDP (S-GDP) |
| | | Population (POP) |
| | Environment | Artificial Surface Rate (ASR) |
| | | Remote-Sensing-based Ecological Index (RSEI) |

As mentioned above, urban location will be reflected by: natural conditions, construction foundation, and policy planning. Location contains many factors. However, many of these factors have the same meaning, which can cause multicollinearity. Two criteria are proposed: (1) The factor should be representative of a facet of location characteristics; (2) The factor should be minimally redundant, but the type should be as detailed as possible [41,56]. We used the LUQ as the Y variable and the location factors as the X variable. The X variables that are not correlated with Y are firstly excluded based on Pearson (*p*-value ≤ 0.1). Then, the variance inflation factor (VIF) analysis was required (VIF < 10). The final 13 drivers, X1 to X13 (Table 2), were selected.

**Table 2.** 13 Factors that passed the Pearson and VIF.

| Type | Name | Description | Pearson | Explanation |
|:---:|:---:|:---:|:---:|:---:|
| Natural Conditions | DEM | Macro-topography | −0.07 *** | Digital Elevation Model; |
| | Relief | Micro-topography | −0.04 ** | DEM variation values within a specific range; |
| | River | Water, Landscape | 0.24 *** | Shali River, overlaps with an urban axis; |
| Construction Foundation | Expressways | Intercity Transportation | −0.09 *** | Includes the Beijing–Hong Kong and Macao Expressway, and the Nanjing–Luoyang Expressway; |
| | Railways | Transportation Materials | 0.09 *** | Mainly includes Beijing–Guangzhou Railway, and Luohe–Fuyang Railway; |
| | High-Speed Railway (HSR) | Transportation People | 0.16 *** | Beijing–Guangzhou High-Speed Railway; |
| | National and Provincial Highways (NPH) | Inner-City Transportation | 0.05 ** | Roads are built and managed by the nation/province. |
| | LUQ$_{2013}$ | Historical Influence | 0.60 *** | Urban growth foundation of the previous stage; |
| | Land use/Land cover (LULC$_{2013}$) | Land Properties | −0.45 *** | Categorical data: Farmland (1); Water (3); Green space (5); Construction (7); |
| Policy Planning | Road Development Axis (RDA) | Structure planning-Main axis | 0.16 *** | Changjiang Road–Renmin Road axis; |
| | General Development Axis (GDA) | Structure planning-Secondary axes | 0.10 *** | Linking administrative, residential, landscape and industrial functional areas; |
| | Industry Cores (IC) | Planning Location | −0.06 *** | The gravity center of the industrial area; |
| | Residential Functional Cores (RFC) | Planning Location | 0.24 *** | The gravity center of the residential area; |

Note: **/*** denote statistically significant at the 5%, 1% level.

Topography was an important factor in modern considerations of the cost of urban development. Additionally, Shali River has historically given additional vitality to this city through the livestock trade relying on water trucking. As development progressed, the artificial network gradually replaced the waterway network. It is important to emphasize that as an L-system, urban is characterized by the output of the previous period as the next period's input. Many factors not only work as a result of the current stage of urban development, but also rejoin the system as drivers of the next stage to establish the cycle. Therefore, LULC and LUQ in 2013 were added for the next growth stage. Regarding policy planning, we focused on spatial structure planning (Table 2). Finally, we have chosen 13 factors and MGWR is employed to assess spatial heterogeneity in the association between each factor and LCI.

## 3. Methods

### 3.1. Assessment Methods

Within the built-up zones, we calculated the RSEI based on Landsat-8; ASR based on LULC; spatial syntax-integration based on roads; spatialized GDP (S-GDP) based on NTL; and combined with population. The weights calculated by criteria importance through intercriteria correlation (CRITIC).

### 3.1.1. Artificial Surface Rate and RSEI

The ASR is obtained from the land use classification above. Compared to traditional indicators, such as green space per capita and forest per capita, the RSEI [57] combines the most intuitive multiple indicators, greenness (NDVI), humidity (WET), dryness (NDBSI), and heat (LST), of the ecological environment. It is a more robust and comprehensive indicator of ecological quality. PCA processed these four indicators, and the first principal component was collected for normalization to obtain the final RSEI:

$$\text{RSEI} = \text{PC1}(\text{NDVI}, \text{WET}, \text{NDBSI}, \text{LST}), \tag{1}$$

### 3.1.2. Integration

Space syntax builds a framework based on the graphical topology. This theory can analyze spatial structures and social systems, as proposed by Bill Hiller [58]. Integration ($I_i$) is a concept specific to spatial syntax and is the reciprocal sum of the shortest paths from one space to another space. It consists of the number of path networks (N) and the topological depth (TD) of this path:

$$I_i = \frac{N\left[\log_2\left(\frac{N+2}{3} - 1\right) + 1\right]}{(N-1)|TD - 1|}, \tag{2}$$

Areas with a high integration are often central locations in urban areas with high pedestrian and vehicular traffic. In this paper, among the many space syntax topology models, we chose the segment model, a refined representation of the axial model, weighting the cost of road 'corners' and incorporating angular properties into the model. This is not available in other road network analyses and is more in line with the "continuous movement mechanism" of pedestrian and vehicular traffic [59].

### 3.1.3. Spatialized GDP

The correction coefficient for total night light (TNL) citywide is based on the GDP data of the secondary and tertiary industry (denoted as $GDP_{23}$) from the statistical document. Here, we tried linear and power function calculations. $DN_i$ is the light intensity value of the i pixel. In this formula: $sGDP_{23}$ is the spatialized $GDP_{23}$, a is the weighting factor for that year, obtained from $GDP_{23}/TNL$. When a linear function is applied, X is the NTL data for the year. When a power function is applied, X is the Ln (NTL+1) data. Pixels without lights are considered to have no secondary or tertiary GDP, so b = 0. The primary industry

GDP ($GDP_1$) is usually generated within farmland. The per-pixel $GDP_1$ was overlaid with $sGDP_{23}$ to obtain sGDP'. The sGDP' zonal statistics were compared with the actual GDP of each county to calculate the error.

$$\text{TNL} = \sum_{i=1}^{n} \text{DN}_i \ (\text{DN}_i \geq 0), \tag{3}$$

$$\text{sGDP}_{23} = a * X + b, \tag{4}$$

The linear correction method is more accurate by comparing mean absolute error (MAE, linear function (%): 19.73; power function (%): 19.87). A further linear correction was made based on the GDP data for each county to obtain the final S-GDP.

### 3.1.4. Curve Fitting of Population

Population datasets are only available up to 2020, and population migration is relatively stable in time series. To ensure data consistency, we constructed a space-time cube to predict the population in 2021. A Curve-fitting model; B Random Forest model; C The exponential smoothing model was chosen. The curve-fitting model is better as it has a lower root mean square error (RMSE) (Table 3). This model chose one of the linear/parabolas/exponentials/Gompertz curve models for one pixel based on the variation of the data over time. We retained the 2020 data to obtain the validation RMSE. Additionally, forecast RMSE indicates the difference between the actual and forecast values for all years.

**Table 3.** Comparison of the accuracy of three methods.

| Type | Forecast RMSE | Validation RMSE |
|---|---|---|
| RMSE of curve-fitting model (People/per pixel) | 0.89 | 0.33 |
| RMSE of random forest model (people/per pixel) | 0.62 | 0.76 |
| RMSE of exponential smoothing model (people/per pixel) | 1.04 | 0.78 |

### 3.1.5. CRITIC Model

The CRITIC model [60] weighted the five indicators above. This weight determination method avoids subjectivity. The final weights $W_j$ are obtained after the following formula:

$$W_j = \frac{C_j}{\sum_{j=1}^{p} C_j}, \tag{5}$$

The CRITIC method is based on multi-criteria decision-making (MCDM). It could ensure the weights by combining the indicator variability $S_j$ (i.e., the degree of dispersion of this indicator), the indicator conflict $R_j$ (i.e., the irrelevance between different indicators), and the information-carrying capacity $C_j$ (obtained by multiplying $S_j$ and $R_j$). These indicator features were calculated within one data group (p group) of the j indicator. It is highly advanced in dealing with covariance and collinearity between the different indicators of urban studies.

### 3.1.6. LUQ Change Intensity

LCI is the rate of LUQ change per year. The extended built-up zone is used as the study boundary:

$$\text{SS} - \text{GII} = \frac{|\Delta I|}{I_0} \times \frac{1}{\Delta T} = \frac{|I_t - I_0|}{I_0} \times \frac{1}{\Delta T}, \tag{6}$$

$I_0$ is the LUQ value of each area before the built-up zone expansion. $I_t$ is the value of each area after the expansion. $\Delta I$ is the change in LUQ value between two time points. $\Delta T$ is the time interval.

*3.2. MGWR Model and Factors Selection*

　　The Y variable, LUQ, was obtained. Exploring the relationship between the Y and X variables requires the multiscale geographically weighted regression (MGWR) model. The MGWR determines specific bandwidth by allowing variables to have different levels of spatial smoothing. Compared with ordinary least squares (OLS) and geographically weighted regression (GWR), it can better calculate the heterogeneity of variables [61].

$$y_i = \sum_{j=1}^{k} \beta_{bwj}(u_i,\ v_i)x_{ij} + \epsilon_i, \tag{7}$$

　　For each sampling point i, MGWR constructs the coordinates $(u_i, v_i)$. $y_i$ is the value of the dependent variable at point i; bwj represents the bandwidth used for the regression coefficient of the j variable. β is the regression coefficient, and $\epsilon_i$ is the random disturbance term. This paper uses the quadratic kernel function and the AICc criterion.

## 4. Results and Analysis

*4.1. Weights and LUQ Assessment*

4.1.1. Subsystem Indicator Weights

　　CRITIC calculates the weights as follows (Table 4). The LUQ is obtained by weighting and overlaying the five factors of the three subsystems.

**Table 4.** The weights in 2013/2017/2021.

| Urban Subsystem | Indicators | Weights | | |
|---|---|---|---|---|
| | | 2013 (%) | 2017 (%) | 2021 (%) |
| SOC | Integration | 23.61 | 22.81 | 22.59 |
| ECO | S-GDP | 21.64 | 19.63 | 18.12 |
| | POP | 10.62 | 9.57 | 9.84 |
| ENV | ASR | 34.53 | 35.28 | 35.16 |
| | RSEI | 9.60 | 12.71 | 14.29 |

4.1.2. Boundary Expansion and LUQ Assessment

　　The results of the built-up zone boundaries and LUQ are shown below (Figure 3 (b1)–(b3)). The built-up zone gradually grows, and small patches die out or merge into the primary landscape. The expansion shows an evident traffic-oriented character, with the region showing a star-shaped pattern along the national and provincial highways (NPH). This might be termed an axial band-like extension pattern [40]. The leading cause of this pattern is increased external traffic. This is a repetition of the aggregation–diffusion–reaggregation cycle. The high values were concentrated in the center and gradually spread along the NPH. The priority expansion of urban space along one or several directions causes a change in the urban morphology [62].

　　The mean values of LUQ in 2013/2017/2021 are 0.51, 0.54, 0.54 respectively, showing a yearly increase but tending to slow down. Based on Figure 3(b1)–(b3), we used the trend analysis tool (ArcGIS 10.8) to draw the trend of each year's assessment value (Figure 4). In the longitude, the quality shows a trend of high in the middle and low around, forming an inverted U curve. Meanwhile, the LUQ in the west is consistently higher than in the east. The trend is similar in latitude, and urban growth is consistently higher in the south than in the north. LUQ has a clear and stable tendency of central aggregation in space.

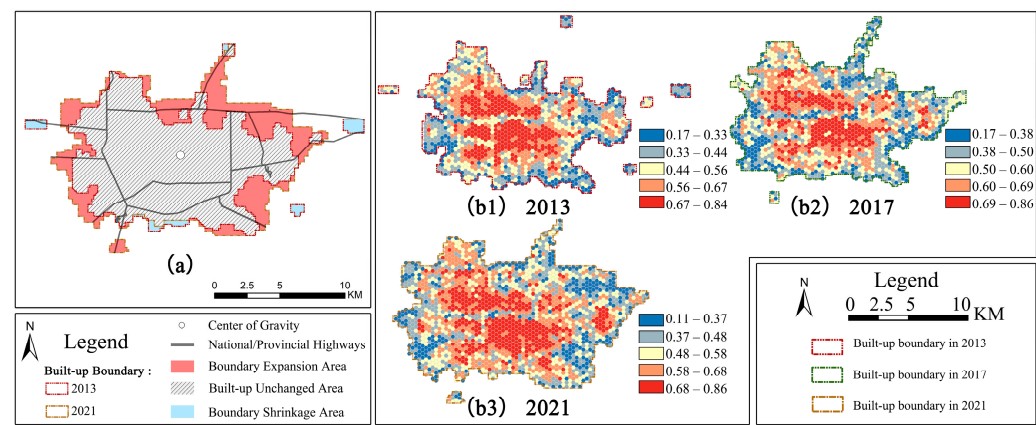

**Figure 3.** (**a**) Diagram of the built-up zone growth–decline area; ((**b1**)–(**b3**)) LUQ in 2013/2017/2021.

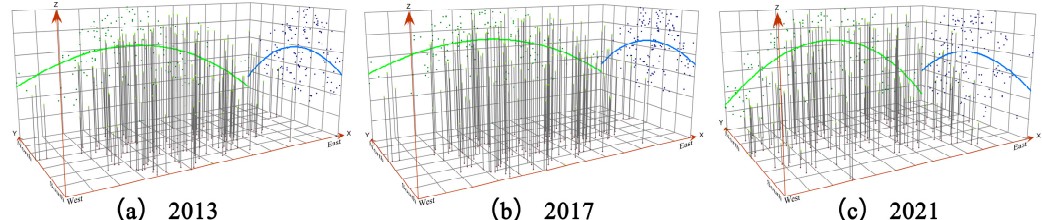

**Figure 4.** Trend analysis of LCI.

### 4.1.3. Spatial Characteristics of LCI

LCI results are as follows (Figure 5(a1)–(a3), Table 5). Compared to the first stage, positive urban growth and degradation became less active between 2017 and 2021. Overall, between 2013 and 2021, the urban core showed degradation ranging from 0 to 6 percent, with new growth poles moving away from the core. From 2013 to 2021, the $LCI_{MEAN} = 0.03$. From 2013 to 2017, the $LCI_{MEAN} = 0.04$, higher than the annual mean over these eight years. From 2017 to 2021, the $LCI_{MEAN} = 0.02$, and growth slowed down. STD was the largest in 2013–2017, with the most unbalanced regional development. Then, STD was reduced in 2017–2021. We found the lowest average STD in 2013–2021 (Table 5). LUQ more equally in space over extended periods.

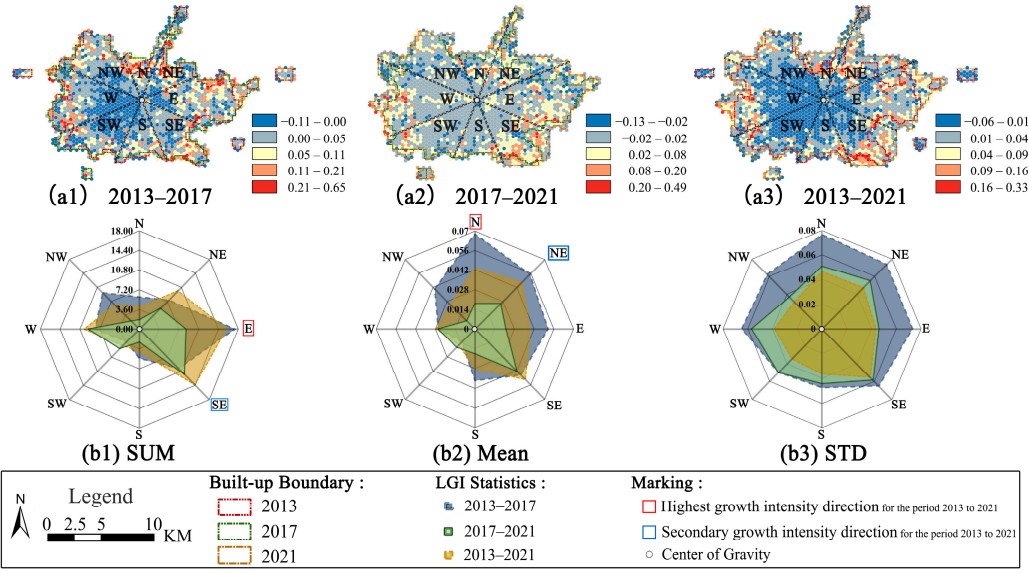

**Figure 5.** (**a1**)–(**a3**) LCI in 3 periods; (**b1**)–(**b3**) Radar chart of LCI zonal statistics.

**Table 5.** LCI statistics' results.

| Statistical Indicators | 2013–2017 | 2017–2021 | 2013–2021 |
|---|---|---|---|
| $LCI_{SUM}$ | 80.01 | 50.41 | 76.08 |
| $LCI_{MEAN}$ | 0.04 | 0.02 | 0.03 |
| $STD$ | 0.07 | 0.05 | 0.04 |

Further zonal statistics are performed on the LCI (Figure 5(b1)–(b3)). In directional analysis, the focus is on three indicators, the sum (SUM), mean (Mean), and standard deviation (STD)of pixel values. The SUM focuses on showing the impact of boundary expansion on LUQ, i.e., the increase in "quantity". The Mean can be understood as an increase in the level of growth, i.e., the increase in "quality". The STD further describes whether this growth occurs evenly over each pixel, in other words, the fairness of the LUQ growth.

We can further observe the urban growth trend to distinguish between expansion and sprawl. For SUM, the LUQ in 2013–2021 is more oriented towards horizontal development, i.e., focusing on the increase in the W/E directions. In sharp contrast, for mean, the N/NE advance together, but later, there is a gradual shift towards the W/SE. It showed an overall vertical-like growth.

Healthy urban development should not only be due to its expansion [63], but the increase in mean should also be more significant than the increase in SUM. When SUM is larger than the mean, such as the E direction, this direction needs to be concerned. Land use in this direction is shifting or will shift to physical sprawl. When the mean is much greater than SUM, consider whether there are factors that limit the expansion of the boundary.

*4.2. Location Factors and Weighted Regression Results*

4.2.1. Location Factors

Based on Table 2, the 13 location factors are spatialized (Figure 6). The river factor in the natural conditions, all transportation factors in the construction foundation, and all policy planning factors were all converted to distance factors.

4.2.2. Calculation Results and Analysis of MGWR

The $R^2$ Adjusted of MGWR was 0.63, which is higher than the OLS of 0.39. The AICc was 4016.23, lower than the OLS of 4749.68, which implies a better fit. The results of MGWR calculations are presented below (Table 6).

**Table 6.** MGWR calculated coefficients and scales for each location factor.

| Variable | Mean | STD | Min | Median | Max | Bandwidth |
|---|---|---|---|---|---|---|
| Intercept | −1.07 | 0.00 | −1.08 | −1.07 | −1.07 | 2022 |
| DEM | −0.07 | 0.04 | −0.11 | −0.08 | 0.00 | 1680 |
| Relief | −0.10 | 0.04 | −0.16 | −0.12 | 0.00 | 925 |
| River | −0.89 | 0.18 | −1.22 | −0.80 | −0.69 | 856 |
| Expressway | 0.74 | 0.00 | 0.74 | 0.74 | 0.75 | 2022 |
| Railways | −0.55 | 0.01 | −0.56 | −0.55 | −0.54 | 2022 |
| High-Speed Railway (HSR) | 1.24 | 1.19 | −3.89 | 1.08 | 4.37 | 44 |
| National/Provincial Highways (NPH) | −0.21 | 0.00 | −0.21 | −0.21 | −0.20 | 2022 |
| Road Development Axis (RDA) | 0.14 | 0.07 | −0.02 | 0.15 | 0.26 | 582 |
| General Development Axis (GDA) | 0.21 | 0.05 | 0.03 | 0.21 | 0.32 | 673 |
| Industrial Cores (IC) | −0.37 | 0.00 | −0.38 | −0.37 | −0.37 | 2022 |
| Residential Functional Cores (RFC) | −0.18 | 0.05 | −0.24 | −0.18 | −0.12 | 1645 |
| $LUQ_{2013}$ | −0.80 | 0.51 | −2.84 | −0.70 | 0.04 | 44 |
| Land Use/Land Cover ($LULC_{2013}$) | 0.04 | 0.03 | −0.01 | 0.04 | 0.08 | 1588 |

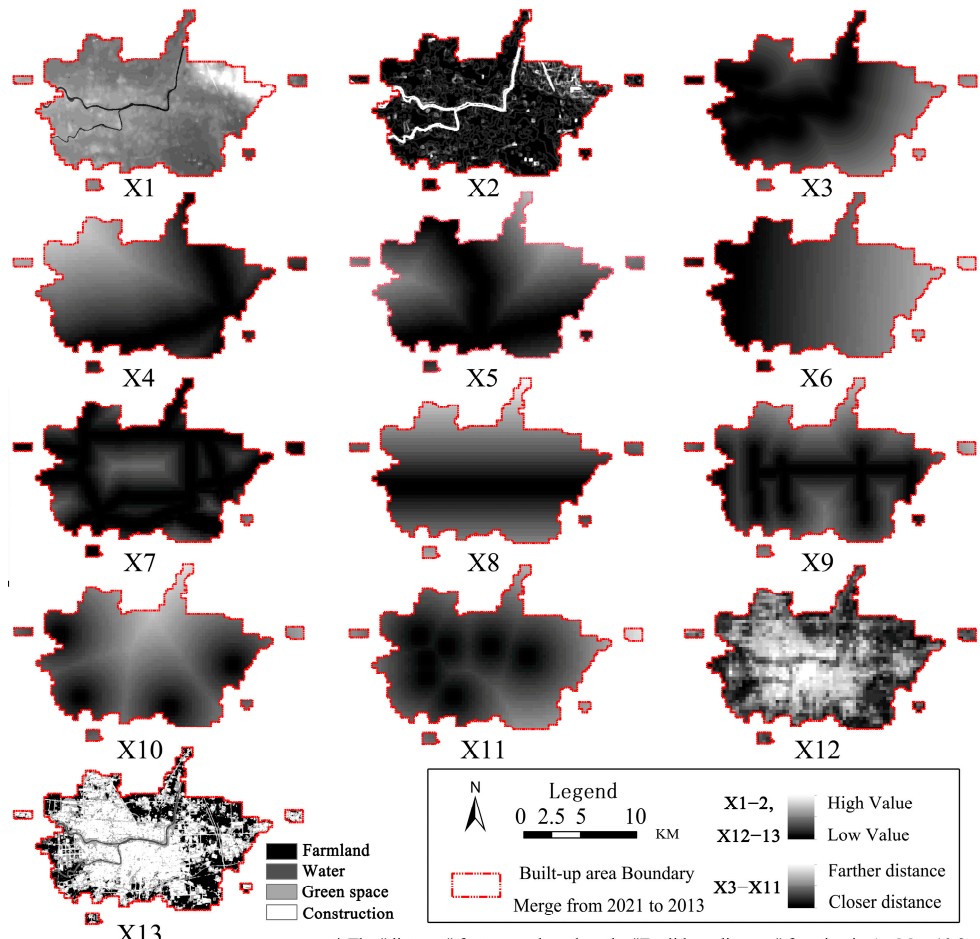

**Figure 6.** (**X1**) DEM; (**X2**) Relief; (**X3**) River; (**X4**) Expressways; (**X5**) Railways; (**X6**) High-speed railway; (**X7**) National/provincial highways; (**X8**) Road development axis; (**X9**) General development axes; (**X10**) Industrial cores; (**X11**) Residential functional cores; (**X12**) $LUQ_{2013}$; (**X13**) Land use/land cover$_{2013}$.

Scale is the most crucial topic in geographic information science [64]. Bandwidth is a direct representation of the action scale and can be understood as the scope to which a factor can exert a uniform influence. The small bandwidth means that different regions are affected by one factor very differently. The HSR, for example, clearly has different importance for the development of different regions, and corresponds to a wide variation of coefficients. In other words, the area over which it can exert a uniform influence is small so the bandwidth is small. A large bandwidth means that a larger area is affected (positively or negatively) by the factor in a relatively uniform way.

This paper controls for nature, transport, spatial structure planning, and historical impact. The intercept indicates the other variables that are not controlled for, such as the culture/cross-city economic attraction. Its bandwidth is 2022, representing around 100% of the total area. The built-up zone merged in 2013 and 2021 is approximately 236.53 km$^2$. Based on Jenks' natural breaks' classification of bandwidth, we have divided these location factors into three categories: ① DEM, expressway, railways, NPH, IC, RFC and LULC are global scales (bandwidth = [1588, 2022]); ② The scales for relief, river, RDA and GDA range from 582 to 925, which are semi-global factors. The scale ranges from 28.75% to 45.70%; ③ HSR and $LUQ_{2013}$ are local factors (bandwidth = 44). Once this scale is exceeded, the coefficient changes dramatically. The semi-global and local factors constitute the concept of 'location'. Their mean bandwidth values are 520.67, accounting for 25.57%.

### 4.2.3. Coefficients' Analysis

The MGWR results presented in the form of pictures can better show the heterogeneous effects of different locus factors on LUQ.

- Semi-global factors

The coefficient changes for the four semi-global factors are shown in Figure 7. Relief represents micro-topographic changes (Table 2), and its influence of it (Mean = −0.1) is more significant than macro-topography (Mean = −0.07). Relief shows a slightly negative effect (Mean = −0.10) and the more complex micro-topography can be a barrier to urban growth. In other words, growth would prefer to occur on flat topography.

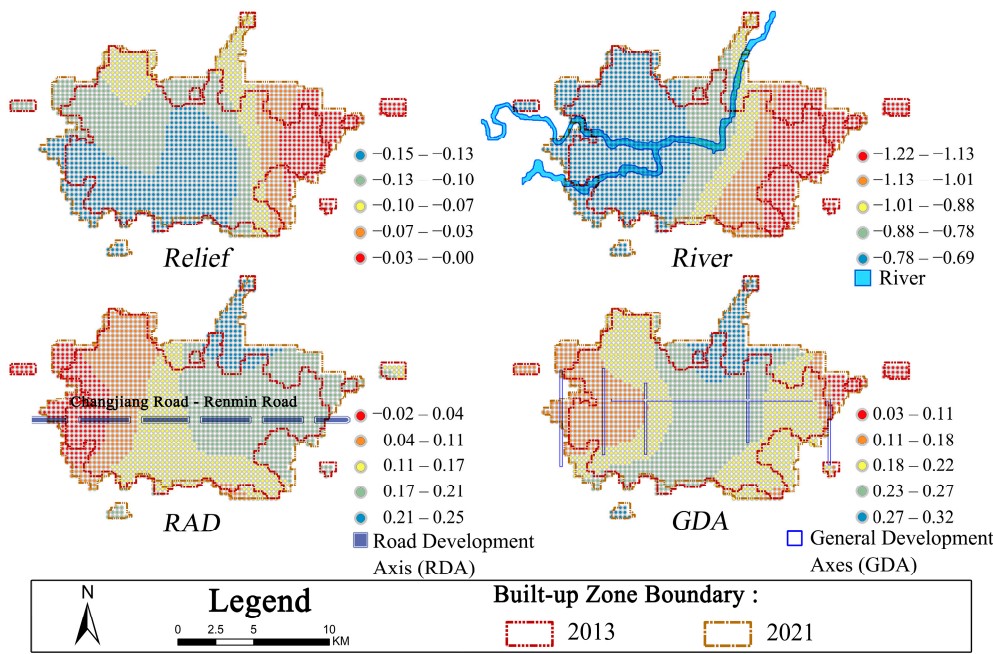

**Figure 7.** Heterogeneity impact of semi-global factors on LCI.

River shows a decisive and overall promotional role, Mean= −0.89. The LUQ decreases by 0.89 percent for each distance unit away from the river. It is easier to boost LUQ in areas near rivers. The construction around the river leads to this heterogeneity. This river has a slightly lower impact on LCI in the west. Moreover, there is a higher attraction to the east.

This paper chooses RAD and GDA as the primary and secondary planning axes for policy planning. RDA's distance coefficient factor was Mean = 0.14. The GDA factor was more significant, and its distance coefficient factor was Mean = 0.21. For each distance unit away from the axis, the LCI increases by 0.14/0.21, separately. It is easier to improve LUQ away from the axes. For RDA, the driving force showed negative impacts in the east, dominated by construction and villages. However, the spatial structure of the urban area still tends to be divided vertically, and the role played by RDA is relatively weak. For the GDA, this is slightly less inhibitory to the periphery and more inhibitory to the center.

- Local factors

The HSR can be considered one of the intercity transportation representatives in the built-up zone (Figure 8). HSR has a distance factor mean coefficient of 1.24. As the only local factor in transport, different regions react to it differently. The primary use of HSR is passenger transport, which is relatively more friendly to the environment. Therefore, the local drive, with the station as the core, has a significant additive effect on the surrounding areas in the south-west region. Combined with LCI$_{2013–2021}$ (Figure 5), this station would be a growth pole in the future.

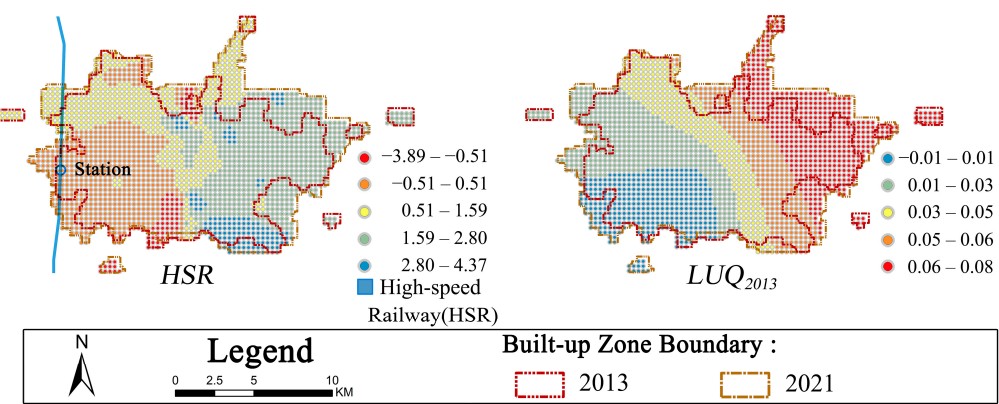

**Figure 8.** Heterogeneity impact of local factors on LCI.

LUQ shows the system foundation in 2013 (Figure 8). There is a negative effect globally (Mean= −0.80), i.e., growth is more likely to be generated in lower-value areas. This is the marginal utility of growth. Overall, it shows a tendency for the coefficients to be lower in the south-west region and higher in the east.

## 5. Discussion

### 5.1. Considerations of LUQ Framework

Due to a certain ambiguity in the LUQ definition, the framework has multiple subsets based on different research objectives. The socio-economic-ecological framework is the most basic and is widely used in fields such as sustainability and land coupling. On this basis, the four-index [7] framework is extended, and even the five-index framework is extended [65]. The environment is the most fundamental physical support for all activities of the subsystems. 'Environment' is often seen as equivalent to nature or ecology. However, the environment is broader and contains a much larger range of substrates, such as grey and green spaces.

Society provides the infrastructure as a bridge between artificiality and nature; society helps to facilitate the spatial flow of materials, energy and information, bridging the urban–rural duality or triadic structure. Usually, society is evaluated using statistics such as local facility density or resource consumption [66,67]. However, obtaining comprehensive and accurate spatial data for developing urban areas is often challenging. This shortcoming becomes more apparent when faced with multi-temporal studies. On the other hand, this needs to consider the changing statistical criterion and temporal heterogeneity of statistical indicators, which usually make studies different across different geographical locations and periods. The road network is often built before the urban system among the many infrastructures. It has a close positive correlation with education/health and care/commerce, and is a necessary prerequisite for running an urban area [68]. Meanwhile, a road network with a solid temporal continuity also avoids doubts about comparability.

### 5.2. Heterogeneity and the Meaning of Scales

In this paper, the differentiation of LUQ is shown in its spatial-temporal non-smoothness. The location impact on LUQ is seen as a combination of homogeneous and heterogeneous location factors. That is, we discuss not only the heterogeneity of location-based LUQ, but also the heterogeneity of the location variables. In previous studies [69], the importance of the drivers is determined by their impact coefficients. We introduce the drivers' bandwidth, extending their filtering dimension. The central idea is that global factors that exert a homogeneous influence on LUQ growth can be considered constant, regardless of their coefficient size. In contrast, semi-global and local factors with a high spatial heterogeneity should be considered as "variables" of greater interest.

In addition to extending the dimension, another advantage of linking heterogeneity to the drivers is that it provides data to support planning at the inner-urban scale. The

locational bandwidth is close to the street scale in an average sense [70]. In this study, excluding the global factors, the mean bandwidth of the semi-global and local factors is 520.67. It works out to an average of 46.86 km$^2$, while the average size of the sub-districts is about 31 km$^2$.

Temporal heterogeneity also needs attention. The formation of urban morphology depends on long time cycles, and different environments influence even the same city at different stages of urbanization, such as with the spillover effect of the planned scope of the urban agglomeration in which it is located and the surrounding urban change, which can then be summed up as a time constraint on urban development [71].

## 6. Conclusions

This paper takes Luohe city as an example, from 2013 to 2021, using the data on ASR, RSEI, integration, spatialized GDP and population within the built-up zones. After weighing the factors through the CRITIC model, a stable framework was constructed to assess LUQ. Afterwards, based on location theory, 13 location factors were selected to explore how they impact the LUQ growth through the MGWR.

LUQ assessment raster display based on subsystem synthesis: (1) The expansion of the built-up zone boundary is an axial band-like extension pattern. LUQ high values clustered in the urban center, and spread outward along the NPH. Elements within the Soc-Eco-Env subsystems are transported outwards along the NPH, advancing the urban star-shaped expansion; (2) LUQ continues to grow, but the growth rate is gradually slowing. Based on the LCI, between 2013 and 2021, the central region saw a slight decline in LUQ, with growth concentrated in the surrounding areas. LCI's zoning statistics show the need to focus on the land quality of the WW/E/SE because land use is more inclined to physical sprawl here.

According to MGWR's measurements: (3) Location factors can be distinguished into global, semi-global, and local factors. The impact of global factors on LUQ is spatially smooth, and they are components of the homogeneous locational space; (4) Similarly, for the growth of LUQ, the semi-global and local factors are components of the heterogeneous locational space. The heterogeneous location factors are ranked in order of influence: HSR > River > LUQ$_{2013}$ > GDA > RDA > Relief.

In the choice of driving factors, public demand for greenness and the ensuing political pressure can create an engine for green development [72,73]. These indicators can be further quantified and added to improve the goodness of fit.

**Author Contributions:** Conceptualization, X.W., H.S. and A.F.; methodology, X.W. and T.B.; software, X.W.; validation, X.W., H.S. and Y.X.; formal analysis, X.W. and Y.X.; investigation, X.Y.; resources, X.Y.; data curation, T.B. and X.Y.; writing—original draft preparation, X.W.; writing—review and editing, L.K., X.W., T.B., H.S., Y.X. and G.T.; visualization, X.W.; supervision, L.K., H.S., Y.X., G.T. and A.F. All authors have read and agreed to the published version of the manuscript.

**Funding:** This research received no external funding.

**Data Availability Statement:** The data presented in this study are available on request from the first author.

**Acknowledgments:** We thank the China Scholarship Council and the Stipendium Hungaricum Programme for supporting some authors' studies and research.

**Conflicts of Interest:** The authors declare no conflict of interest.

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
