# Peer review of "Land Use Quality Assessment and Exploration of the Driving Forces Based on Location: A Case Study in Luohe City, China"

_land, doi:10.3390/land12010257_

Round 1

Reviewer 1 Report

Article title:

"Land Use Quality Assessment and Exploration of the driving mechanism Based on Location: A Case Study in Luohe, China"

Keywords:

Land Use Quality; Driving mechanism; Society-Economy-Environment; Weighted regression; Location Theory

Comments:

The research is of particular interest because of the focus on an objective methodology for assessing land use, which is an important tool for urban planners. In terms of content, the research combines many important data to reach indicators that help to understand the best areas for urban expansion. A good LUQ assessment can really help people plan, develop, and use land resources in a more scientific and sensible way.

The abstract is well written and reflects the content of the research in terms of the research issue, methodology, and results.

Introduction generally speaking, the literature review needs to be developed.

L. 48 “the city could be considered a complex system” it is a fundamental issue in such research and therefore this part should be better explained.

L. 53 needs a reference

L. 60 “Urban location theory” In this part of the research, reference should be made to the works of the Chicago School of thought (Park & Burgess, 1925, Hoyt 1939, Ullman 1945).

The rest of the research seems acceptable.

Reviewer 2 Report

The authors have done a lot of research, but some wishes need to be made.

It is not entirely clear how the encroachment of urbanization directly leads to serious problems for urban sustainability, and low density has led to sprawl (introduction). If the direction of the research is aimed only at the advantages of compact urban development, then this must be indicated at the beginning of the research. It would be desirable for the authors to express their point of view.

The authors note, «According to the above discussion, the location factors are finally decomposed into three main parts: natural conditions, construction foundation, and policy planning» (Line 84). However, in the previous discussion to which the authors refer, there was no mention of the impact of natural conditions and planning policy on the heterogeneity. It is desirable that the authors substantiate in detail why these three elements were chosen for further research.

The land use quality is a multifactorial indicator of suitable for a particular use. The key elements of the study are the assessment of Land Use Quality (LUG) and Land Use Quality Growth Intensity (LGI). The article indicates LGI is an index determined by calculating the annual rate of LUQ change, and it can help us better focus on LUQ growth (Line 86-88). However, quality can be assessed both positively and negatively. Perhaps here it is advisable to consider the modification of quality and not it growth. To avoid misconception, it would be desirable to clarify exactly of semantic content of concepts "Land Use Quality" and «Land Use Quality Growth Intensity" in these studies.

The heterogeneity of urban space is the result of the cross-influence of socio-economic, infrastructural, cultural and other factors. The authors note this (lines 79-80). However, in "Construction Foundation" (Table 2) mainly transport routes are investigated. It would be interesting what other factors were considered and why they were excluded.

The choice of indicators to describe the Urban Subsystems (Society, Economy, Environment) is not entirely clear. It is desirable to explain the ideology behind the choice of indicators. 

The conclusions do not reflect and systematize all the results of the study. 

General recommendations. Adequate research results, especially in the modeling part, directly depend on the validity of the chosen prerequisites. The authors should pay more attention to the scientific substantiation of the arguments and indicators on which the research is based.

Reviewer 3 Report

The paper studied the urban land use quality change in Luohe, China. There is a significant amount of spatial analysis work being done by the paper. The spatial analysis results are presented in maps and charts with great quality. However, the structure of the paper is hard to follow and needs to be corrected before being considered for publishing.

1) The results in the abstract section need to focus on the findings of this paper instead of making policy assumptions. Such as "planners should focus on". The authors need to clarify what are the real findings of this study

2) the first section needs to be broken into an Introduction section and a literature review section. The introduction needs to define the study questions and goals of the study clearly. Key terms need to be defined and explained, such as what is the land use quality means. why is it important? 

3) It is recommended to bring the study framework Figure 2 up to the top of the study area and data section so the readers can understand what the paper is trying to do and why the data is needed

4) there should be some background information on the study area, the population, the land use change, and other key characteristics to justify why Luohe was chosen for the study.

5) some of the abbreviations need to be defined first before use. For instance, MGWR, UGB and so on. The authors need to check the order of the abbreviations.

6) the discussion section needs to focus on the results and justify the findings. The authors need to clarify if global, semi-global and local divisions are method or results.

7) the language of the paper needs to be checked. Some of the sentences do not make sense, such as "secondary residential prices" "converge in the main urban", and so on.

Round 2

Reviewer 3 Report

The authors have made significant improvements to the manuscript. The issues raised in the first round of review have been addressed. It is recommended to publish. Great work.